# $DA^2$-VPR: Dynamic Architecture for Domain-Aware Visual Place Recognition

## Abstract

Visual Place Recognition (VPR) systems struggle with training-to-test domain shifts caused by environmental changes such as lighting, weather, and seasonal variations. Existing methods rely on input-invariant strategies with fixed parameters, which restrict their ability to cope with diverse test conditions. We propose Dynamic Architecture for Domain Aware Visual Place Recognition ($DA^2$-VPR), a dynamic feature modulation framework that adapts representations according to input scene characteristics. By dynamically modulating features across spatial and channel dimensions using foundation model features as conditioning signals, our method effectively narrows the training-to-testing gap. Our framework consists of: (1) a dynamic adapter that adjusts representations to scene conditions, (2) a transformer aggregator with adaptive query generation from input features, and (3) domain-variance augmentation with texture and appearance modifications. Experiments on challenging VPR benchmarks with significant domain shifts show that $DA^2$-VPR consistently outperforms input-invariant baselines, demonstrating superior generalization and establishing new state-of-the-art results.

## 1 Introduction

Visual Place Recognition (VPR) aims to identify the same location from a query image within a reference database. It plays a crucial role in real-world applications such as autonomous driving (Chen et al., 2023), robot navigation (Chen et al.; Hausler et al., 2019), and augmented reality (Garg et al., 2021). As VPR operates solely on 2D image inputs without external sensors (e.g., LiDAR, GPS), it must be robust to diverse visual variations, such as lighting, weather, season, viewpoint, and sensor differences.

Recent VPR methods (Izquierdo & Civera, 2024; Lu et al., 2024d; Jin et al., 2025; Qiu et al., 2024) typically train on urban imagery (Ali-bey et al., 2022) and evaluate on distinct domains to test generalization. These approaches adopt a two-stage pipeline: a feature extractor encodes the image, and an aggregator summarizes local features into a global descriptor. While many studies have explored improved aggregation strategies (Lu et al., 2024c;b; Jin et al., 2025), most efforts have focused on addressing domain variations primarily through enhanced aggregator designs, with limited exploration of domain-aware feature adaptation that dynamically modulates learnable parameters based on input scene characteristics.

However, despite these advances in aggregation strategies, a critical challenge remains underexplored in the form of training-to-test domain shifts. These shifts arise from changes in geography, lighting conditions, seasonal variations, or sensor characteristics between training and test environments. Such cross-domain generalization challenges pose significant difficulties to VPR systems, as they can dramatically degrade performance despite strong training performance. To systematically address these domain shift challenges, we identify that VPR faces two distinct types of domain gaps: the inter-task gap, which stems from representational mismatch between pretraining objectives (e.g., classification) and the VPR retrieval task; and the inter-dataset gap, which results from environmental variations between training and test conditions.

While existing approaches have primarily addressed the inter-task gap through input-invariant adaptation methods such as fine-tuning or static adapters on foundation models, the inter-dataset gap remains largely unresolved. Current methods rely on input-invariant adaptation strategies that main-

Figure 1: **Overview of domain-adaptive tuning strategies for VPR.** To adapt pretrained backbones, existing methods (a) fine-tune partial layers or (b) add static adapters before a Transformer generates global descriptors. (c) Our approach uses dynamic tuning to improve robustness against domain shifts. Nordland samples illustrate extreme seasonal changes, the map shows query-reference matches with a highlighted building.

tain fixed parameter configurations throughout inference, limiting their ability to handle diverse environmental conditions at test time. As illustrated in Figure 1, these approaches either (a) fine-tune partial backbone layers with predetermined structures, or (b) insert static adapters with fixed parameters that cannot adjust to varying input conditions. Although recent efforts such as EMVP (Qiu et al., 2024) have begun to consider domain bias in spatial semantics (e.g., urban vs. rural), these methods still employ static adaptations that lack the flexibility to dynamically respond to scene-specific characteristics during inference.

The limitation of input-invariant adaptation becomes particularly evident when dealing with diverse environmental conditions. For instance, features that are discriminative for place recognition in daytime urban scenes may become less effective in nighttime or rural environments. This motivates the need for dynamic adaptation mechanisms that can modulate feature representations based on the specific characteristics of input scenes, as demonstrated by the challenging examples from the Nordland (Olid et al., 2018) dataset in Figure 1(c).

To enable such dynamic adaptation capabilities, we propose a dynamic feature modulation framework for VPR that adapts representations based on scene-specific characteristics. Our approach draws inspiration from recent advances in scene-aware adaptation for dense prediction tasks (Zhou et al., 2021), where dynamic modulation has shown effectiveness in handling scene-level variations. Although VPR and dense prediction tasks differ in their supervision signals and objectives, both require robustness to scene-level variations such as viewpoint changes, illumination conditions, and structural differences. This shared requirement for scene adaptability motivates our design of a dynamic modulation architecture specifically tailored for the VPR retrieval task.

Our $\underline{D}$ynamic $\underline{A}$rchitecture for $\underline{D}$omain $\underline{A}$ware Visual Place Recognition ($DA^2$-VPR) consists of three key components designed to enable scene-aware feature modulation. First, we introduce a dynamic adapter that modulates features across both spatial and channel dimensions, conditioned on the features extracted from the foundation model feature extractor. This enables the model to adaptively adjust its internal representations to diverse environmental conditions such as changes in lighting, season, or viewpoint. Second, we design a transformer-based aggregator with scene-aware attention mechanisms that dynamically generate query embeddings from the input features themselves, allowing for adaptive aggregation of salient local features unlike prior methods that rely on fixed, learnable queries. Third, we propose a domain-variance augmentation strategy that extends beyond conventional geometric perturbations by applying augmentations such as texture variations and appearance modifications during training, enhancing the model's robustness to domain variations and improving generalization capabilities.

We validate our method on challenging VPR benchmarks that involve significant domain shifts, including day/night transitions and seasonal changes. Our experimental results demonstrate consistent improvements over input-invariant adaptation baselines, confirming the effectiveness of dynamic modulation in achieving robust generalization across diverse visual conditions.

Our contributions are summarized as follows:

- To the best of our knowledge, this is the first VPR framework that dynamically modulates features based on scene-adaptive embeddings from a foundation model, addressing the limitations of input-invariant adaptation.

- We propose a transformer aggregator that dynamically generates query embeddings conditioned on the input, enabling scene-aware attention-based feature aggregation.

- We empirically demonstrate the effectiveness of our method on challenging VPR benchmarks with significant domain shifts, showing strong generalization under diverse visual conditions.

## 2 RELATED WORKS

### 2.1 VISUAL PLACE RECOGNITION

Visual Place Recognition (VPR) localizes a query image by retrieving the most similar reference from a large-scale database, requiring robustness to illumination, seasonal, and viewpoint variations.

Traditional methods used hand-crafted local features such as SIFT (Lowe, 2004) and SURF (Bay et al., 2006), aggregated with BoW (Zhang et al., 2010), VLAD (Jégou et al., 2010), or Fisher Vector (Perronnin & Dance, 2007). While efficient, they were highly sensitive to environmental changes.

Deep learning approaches improved robustness through feature learning and aggregation. NetVLAD (Arandjelovic et al., 2016) and its extensions (e.g., VLAD-BuFF (Khaliq et al., 2024), SuperVLAD (Lu et al., 2024d), EMVP (Qiu et al., 2024)) addressed feature burstiness, clustering limitations, and inductive bias via advanced normalization and centroid-free strategies.

Recently, transformer-based models introduced query-driven aggregation. BoQ (Ali-Bey et al., 2024) applies cross-attention queries for global probing, and EDTFormer (Jin et al., 2025) enhances descriptors via decoder-based context modeling.

Despite these advances, most methods remain constrained by the domain sensitivity of backbone features. Thus, robust VPR requires not only advanced aggregation but also input-adaptive feature refinement using scene-specific statistics.

### 2.2 PARAMETER EFFICIENT FINE TUNING FOR VPR

Following AnyLoc (Keetha et al., 2023), which showed strong zero-shot VPR performance using classical aggregation methods (e.g., NetVLAD (Arandjelovic et al., 2016), GeM (Radenović et al., 2018)) on top of Visual Foundation Models (VFMs), recent works leverage pre-trained models like DINO (Caron et al., 2021; Oquab et al., 2024) and SAM (Kirillov et al., 2023) for VPR (Keetha et al., 2023; Garg et al., 2024). While these models provide powerful general-purpose features, direct transfer to VPR is limited by task-specific challenges.

To mitigate this, adaptation strategies include full or partial fine-tuning (Lu et al., 2024a; Izquierdo & Civera, 2024; Ali-Bey et al., 2024; Lu et al., 2024d) and parameter-efficient fine-tuning (PEFT) via lightweight adapter modules (Lu et al., 2024c;b; Jin et al., 2025; Qiu et al., 2024). Existing PEFT methods efficiently train small modules but are mostly static, lacking flexibility to handle input- or domain-level variations. EMVP (Qiu et al., 2024) partially addresses this with Dynamic Power Normalization (DPN), yet dynamic modulation of the feature extraction process remains limited.

### 2.3 DYNAMIC MODULATION

Dynamic feature modulation enhances robustness to domain shifts by adapting representations to input characteristics. Prior works such as HyperNetworks (Ha et al., 2017), Dynamic Filter Networks (Jia et al., 2016), and Dynamic Decoupled Filter (DDF) (Zhou et al., 2021) dynamically generate filters or weights, enabling flexible adjustment along spatial and channel dimensions. Building on these ideas, we design an input-aware dynamic adapter tailored for VPR to produce domain-robust features.

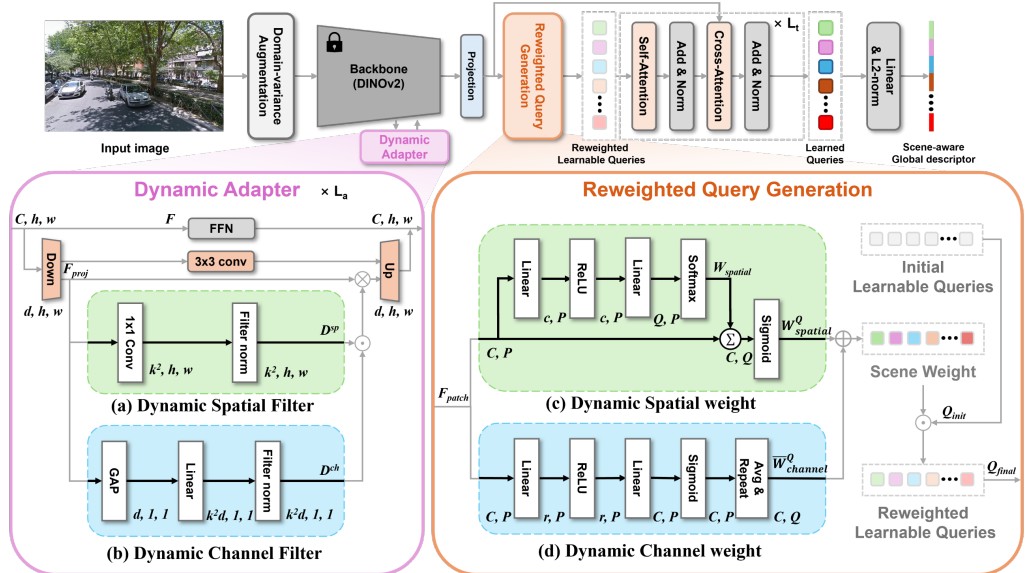

Figure 2: **Overall pipeline of $DA^2$-VPR.** Scene-aware features are extracted via a Dynamic Adapter inserted into the frozen DINOv2 backbone. These dynamically adapted features are then used to generate spatially and channel-wise reweighted learnable queries for an efficient transformer aggregator. The dynamic adapter modulates intermediate representations based on scene-specific visual characteristics, while the adaptive query weights focus attention on informative spatial regions and channel dimensions, enabling robust global descriptor generation for diverse visual conditions.

## 2.4 QUERY ADJUSTMENT

Transformer-based VPR models (Ali-Bey et al., 2024; Jin et al., 2025) utilize learnable queries, but most employ fixed latent vectors independent of input conditions. Recent advances, such as Conditional DETR (Meng et al., 2021; Chen et al., 2022), Anchor-DETR (Wang et al., 2022b), and UniAS (Ma et al., 2025), demonstrate the benefit of input- or task-dependent query adjustment for more expressive attention. Inspired by these approaches, we introduce an input-adaptive query enhancement designed for robust VPR.

## 3 METHOD

The overall $DA^2$-VPR pipeline, as illustrated in Figure 2, adopts a similar structure to existing approaches (Ali-Bey et al., 2024; Jin et al., 2025) that use DINOv2 as the feature extractor and a transformer architecture as the descriptor aggregator. To effectively address the domain gap in VPR, the backbone is tuned with a dynamic adapter structure (Section 3.1). Subsequently, the tuned features obtain reweighted learnable queries through reweighted query generation (Section 3.2).

### 3.1 FEATURE EXTRACTOR WITH DYNAMIC ADAPTER

This study proposes a Dynamic Adapter module that modulates feature extraction dynamically based on the visual characteristics of the input image. As illustrated in Figure 2, the adapter is inserted in parallel with the Feed Forward Network (FFN) layers of the DINOv2 backbone, specifically within the last $L_a$ layers. It consists of two lightweight, independently operating sub-modules that inspired by the Dynamic Decoupled Filter (Zhou et al., 2021) framework: (a) Dynamic Spatial Filter and a (b) Dynamic Channel Filter.

Given an intermediate feature map from the backbone $F \in \mathbb{R}^{C \times h \times w}$, we first apply a linear projection along the channel dimension to obtain a projected feature map $F_{proj} \in \mathbb{R}^{d \times h \times w}$, where $d < C$. To capture both fixed local structures and input-dependent variations, $F_{proj}$ is processed through two parallel branches: a learnable $3 \times 3$ static convolution that enhances local spatial con-

text, and *the dynamic adapter, which generates spatially and channel-wise adaptive kernels based on the input features.* This parallel design enables the network to leverage both static local patterns and scene-adaptive modulations for robust feature extraction.

### 3.1.1 DYNAMIC FILTER GENERATION

To adaptively modulate features along both spatial and channel dimensions, the dynamic adapter generates two types of filters from the projected feature map $F_{proj} \in \mathbb{R}^{d \times h \times w}$.

**Dynamic Spatial Filter.** The dynamic spatial filter generates adaptive spatial kernels for each spatial location. Specifically, from the feature $F_{proj}$, we use a $1 \times 1$ convolution to dynamically generate spatial filter weight at each position:

$$D^{sp} = \text{FN}(\text{Conv}_{1\times1}(F_{proj})) \in \mathbb{R}^{k^2 \times h \times w} \ , \tag{1}$$

where $k$ denotes the spatial kernel size, and $\text{FN}(\cdot)$ is a filter normalization function that stabilizes the dynamically generated kernels (see Supplementary Material A.1 for details).

**Dynamic Channel Filter.** The dynamic channel filter produces channel-wise adaptive kernels based on global feature statistics. We first extract a global descriptor using global average pooling (GAP) over the spatial dimensions and then transform it via a lightweight MLP followed by normalization:

$$D^{ch} = \text{FN}\left(\text{MLP}(\text{GAP}(F_{proj}))\right) \in \mathbb{R}^{k^2 d \times 1 \times 1}. \tag{2}$$

This allows each channel to be modulated according to the global scene characteristics of the input. By jointly leveraging $D^{sp}$ and $D^{ch}$, the adapter modulates features independently along spatial and channel dimensions, capturing both local variations and global dependencies in a complementary manner.

### 3.1.2 FEATURE MODULATION

Given the dynamically generated spatial and channel filters, the final modulated feature at channel $r$ and spatial location $i$ is computed by aggregating neighboring features within the $k \times k$ local region:

$$\hat{F}(r, i) = \sum_{j \in \Omega(i)} D_i^{sp}[p_i - p_j] \cdot D_r^{ch}[p_i - p_j] \cdot F_{proj}(r, j) \ , \tag{3}$$

where $\Omega(i)$ denotes the local neighborhood of position $i$, and $p_i - p_j$ represents the relative spatial offset within the $k \times k$ kernel, i.e., $[p_i - p_j] \in \{(-\frac{k-1}{2}, -\frac{k-1}{2}), \ldots, (\frac{k-1}{2}, \frac{k-1}{2})\}$, with $p_i$ denoting the 2D coordinates of pixel $i$. Here, $\hat{F}(r, i) \in \mathbb{R}$ denotes the output feature value at the $i$-th pixel and $r$-th channel, while $F_{proj}(r, j) \in \mathbb{R}$ is the input feature value at the $j$-th pixel and $r$-th channel. The spatial filter is defined as $D^{sp} \in \mathbb{R}^{k^2 \times h \times w}$, where $D_i^{sp} \in \mathbb{R}^{k \times k}$ denotes the location-specific kernel at pixel $i$. Similarly, the channel filter is defined as $D^{ch} \in \mathbb{R}^{k^2 d \times 1 \times 1}$, where $D_r^{ch} \in \mathbb{R}^{k \times k}$ denotes the channel-specific kernel at channel $r$. This formulation clearly distinguishes between the full dynamic filter tensors $(D^{sp}, D^{ch})$ and their pixel- or channel-specific instances $(D_i^{sp}, D_r^{ch})$, ensuring precise interpretation.

## 3.2 REWEIGHTED QUERY GENERATION

We extend the learnable query design in DETR-based architectures (Wang et al., 2022b; Meng et al., 2021; Chen et al., 2022) by introducing a reweighted query generation mechanism. Unlike fixed queries, our approach adaptively modulates query embeddings according to both spatial relevance and channel importance, thereby improving robustness against scene variations in VPR.

As illustrated in Figure 2(c), given the modulated feature map $F \in \mathbb{R}^{C \times h \times w}$ from the Dynamic Adapter, we divide it into non-overlapping patches of size $p \times p$ and reshape them into flattened tokens:

$$F_{patch} \in \mathbb{R}^{C \times P}, \quad P = \frac{h \times w}{p^2}. \tag{4}$$

**Dynamic Spatial Attention Weight.** To highlight spatially relevant regions, we compute spatial attention weights via a lightweight two-layer MLP followed by a softmax activation:

$$W_{spatial} = \text{Softmax}(MLP(F_{patch})) \in \mathbb{R}^{Q \times P}. \tag{5}$$

where $Q$ denotes the number of queries. The spatial contribution for each query is then obtained by projecting $F_{patch}$ onto $W_{spatial}$:

$$W_{spatial}^{Q} = \sigma \left( F_{patch} \cdot W_{spatial}^{\top} \right) \in \mathbb{R}^{C \times Q}, \tag{6}$$

where $\sigma(\cdot)$ represents the sigmoid activation.

**Dynamic Channel Attention Weight.** To adaptively emphasize informative channels, channel attention weights are generated with another two-layer MLP followed by a sigmoid:

$$W_{channel} = \sigma(MLP(F_{patch})) \in \mathbb{R}^{C \times P}. \tag{7}$$

These weights are spatially averaged to form a global channel descriptor, which is then broadcasted to all queries:

$$\bar{W}_{channel}^{Q} \in \mathbb{R}^{C \times Q}. \tag{8}$$

Finally, the reweighted queries are formed by combining spatially and channel-modulated weights, providing both local adaptivity and global context awareness.

### 3.2.1 REWEIGHTED LEARNABLE QUERIES

Finally, the reweighted learnable queries are derived by combining both spatial and channel modulations with the initial learnable queries $Q_{init} \in \mathbb{R}^{C \times Q}$:

$$Q_{final} = Q_{init} \odot (W_{spatial}^{Q} + \bar{W}_{channel}^{Q}), \tag{9}$$

where $\odot$ denotes element-wise multiplication.

By dynamically modulating queries according to spatial relevance and channel importance conditioned on input-specific characteristics, the proposed adaptive mechanism can achieve improved representational flexibility compared to fixed query approaches.

### 3.3 ROBUST TRAINING STRATEGY WITH AUGMENTATIONS

To effectively supervised our dynamic architecture, We adopt the Multi-Similarity (MS) loss (Wang et al., 2019), which explicitly optimizes fine-grained similarity relationship among positive and negative pairs within each training batch. Given L2-normalized embeddings $\mathbf{f}_i$ and $\mathbf{f}_j$ for samples $i$ and $j$, their cosine similarity is defined as $S_{ij} = \mathbf{f}_i^{\top} \mathbf{f}_j$. Positive pairs $\mathcal{P}(i)$ are generated as augmented views of the same image, while negative pairs $\mathcal{N}(i)$ originate from different images. The MS loss simultaneously pulls positive closer and pushes apart hard negatives based on a predefined similarity margin $\lambda$. Formally, the MS loss is expressed as:

$$\mathcal{L}_{MS} = \frac{1}{m} \sum_{i=1}^{m} \left\{ \frac{1}{\alpha} \log \left[ 1 + \sum_{k \in \mathcal{P}_i} e^{-\alpha(S_{ik} - \lambda)} \right] + \frac{1}{\beta} \log \left[ 1 + \sum_{k \in \mathcal{N}_i} e^{\beta(S_{ik} - \lambda)} \right] \right\}, \tag{10}$$

where $\alpha$ and $\beta$ control the gradient scaling, and the margin $\lambda$ determines the threshold for hard sample mining. This approach effectively encourages learning discriminative and robust feature representations.

While the original MS loss typically employs standard augmentations, including geometric transformations (identity, shear, translation, rotation) and photometric adjustments (brightness, color jittering, contrast, sharpness), these alone are insufficient to address the significant domain shifts encountered in VPR scenarios.

Table 1: Comparison of Recall@k (%) on multiple benchmark datasets with pronounced domain variations. The best is highlighted in **bold** and the second is underlined, and "–" indicates values not reported. In the table, Eyn., S.N. and S.S. denote Eynsham, SVOX Night and SVOX Snow respectively. The backbone column abbreviations R, B, and L indicate ResNet-50, DINOv2-B, and DINOv2-L, respectively. BoQ[†] indicates the model trained with an image size of 224 for fair comparison with other DINO variants.

| Method | Back bone | Nordland* | | | Nordland** | | | AmsterTime | | | Eyn. | S. N. | S. S. |
|---|---|---|---|---|---|---|---|---|---|---|---|---|---|
| | | R@1 | R@5 | R@10 | R@1 | R@5 | R@10 | R@1 | R@5 | R@10 | R@1 | R@1 | R@1 |
| MixVPR | R | 58.4 | 74.6 | 80.0 | 76.2 | 86.9 | 90.3 | 40.2 | 59.1 | 64.6 | 86.6 | 64.4 | 96.8 |
| EigenPlace | R | – | – | – | 71.2 | 83.8 | 88.1 | 48.9 | 69.5 | 76.0 | 90.7 | 58.9 | 93.1 |
| BoQ | R | 70.7 | 84.0 | 87.5 | 83.1 | 91.0 | 93.5 | 52.2 | 72.5 | 78.4 | 91.3 | 87.1 | 98.7 |
| BoQ[†] | B | 77.5 | 89.8 | 92.9 | 87.4 | 94.8 | 96.7 | 61.8 | 82.3 | 86.3 | 91.9 | 96.5 | 98.3 |
| SALAD | B | 76.0 | 89.2 | 92.0 | 89.7 | 95.5 | 97.4 | 58.8 | 79.0 | 84.2 | 91.6 | 95.4 | 98.9 |
| VLAD-BuFF | B | 73.4 | 88.4 | 91.5 | 85.1 | 93.8 | 96.0 | 59.0 | 78.5 | 83.6 | 91.6 | 95.5 | 98.7 |
| EDTFormer | B | 73.1 | 86.7 | 90.1 | 88.3 | 95.3 | 97.0 | 65.2 | 85.0 | 89.0 | 92.1 | 96.2 | 98.7 |
| EMVP | L | 78.4 | 89.7 | 92.4 | 88.7 | 97.3 | **99.3** | – | – | – | – | – | – |
| $DA^2$-VPR-B | B | 83.4 | 93.5 | 95.4 | 93.3 | 97.7 | 98.6 | 63.7 | 84.5 | 87.3 | 92.4 | 97.7 | **99.0** |
| $DA^2$-VPR-L | L | **86.4** | **95.3** | **96.9** | **95.4** | **98.4** | 99.1 | **66.9** | 85.3 | 89.2 | 92.6 | 98.5 | 99.0 |

Table 2: Comparison of recall@k (%) on standard benchmark datasets. the best is highlighted in **bold** and the second best is underlined, and "–" indicates values not reported.

| Method | Back bone | Pitts250k | | | Tokyo24/7 | | | MSLS-val | | | SPED | | |
|---|---|---|---|---|---|---|---|---|---|---|---|---|---|
| | | R@1 | R@5 | R@10 | R@1 | R@5 | R@10 | R@1 | R@5 | R@10 | R@1 | R@5 | R@10 |
| NetVLAD | R | 90.5 | 96.2 | 97.4 | 60.6 | 68.9 | 74.6 | 82.6 | 89.6 | 92.0 | 78.7 | 88.3 | 91.4 |
| MixVPR | R | 94.6 | 98.3 | 99.0 | 85.1 | 91.7 | 94.3 | 88.0 | 92.7 | 94.6 | 85.2 | 92.1 | 94.6 |
| EigenPlace | R | 94.1 | 97.9 | 98.7 | 93.0 | 96.2 | 97.5 | 89.1 | 93.8 | 95.0 | 70.2 | 83.5 | 87.5 |
| BoQ | R | 95.0 | 98.5 | 99.1 | 94.3 | 96.5 | 96.5 | 91.2 | 95.3 | 96.1 | 86.5 | 93.4 | 95.7 |
| BoQ[†] | B | 96.0 | 98.9 | 99.3 | 96.9 | 98.7 | 99.0 | 92.4 | 96.2 | 96.9 | 92.2 | 95.7 | 96.4 |
| SALAD | B | 95.1 | 98.5 | 99.1 | 94.6 | 97.5 | 97.8 | 92.2 | 96.4 | 97.0 | 92.1 | 96.2 | 96.5 |
| VLAD-BuFF | B | 95.5 | 98.5 | 99.2 | 96.2 | 98.7 | **99.4** | 91.8 | 96.0 | 96.2 | 91.4 | 95.9 | 96.9 |
| EDTFormer | B | 95.9 | 98.8 | 99.3 | 97.1 | 98.1 | 98.4 | 92.0 | 96.6 | 97.2 | 92.4 | 95.9 | 96.9 |
| EMVP | L | 96.5 | 99.1 | **99.5** | – | – | – | **93.9** | 97.3 | 97.6 | **94.6** | 97.5 | 98.4 |
| $DA^2$-VPR-B | B | 96.2 | 98.9 | 99.3 | 97.1 | **99.1** | **99.4** | 93.2 | 96.8 | 96.9 | 92.4 | 95.9 | 96.7 |
| $DA^2$-VPR-L | L | **97.0** | **99.3** | **99.5** | **97.8** | 98.7 | **99.4** | 93.5 | 97.3 | 97.7 | 93.6 | 97.5 | 98.5 |

Therefore, to explicitly bridge the domain gap, we introduce additional synthetic degradations inspired by RobustSAM (Chen et al., 2024). Specifically, we augment the training data with simulated gamma contrast, weather conditions (fog, snow, rain) via realistic texture overlays, as well as structural corruptions (cutout, motion blur and perspective transforms) to represent occlusions and distortions commonly observed in real-world VPR tasks. By exposing our dynamic modules-the Dynamic Adapter and Reweighted Query Generation-to these more diverse and challenging scenarios during training, we aim to encourage them to learn representations that are better able to adapt to varying environmental conditions encountered at inference time.

## 4 EXPERIMENTS

In this section, we first describe the datasets and implementation details. We then provide extensive comparisons with state-of-the-art methods and detailed ablation studies validating our design choices.

### 4.1 DATASETS

We evaluate the effectiveness and domain generalization performance or our method using ten challenging VPR benchmark datasets, each featuring significant visual variations including seasonal, illumination, weather, viewpoint, and color changes. These datasets comprehensively represent real-world generalization capabilities. Detailed dataset configurations and evaluation protocols are

described in the Supplementary Materials (B). For training, we use GSV-Cities dataset (Ali-bey et al., 2022), a large-scale dataset consisting of diverse urban street-view images. This dataset has been widely adopted in recent VPR studies for learning generalizable visual representations.

## 4.2 IMPLEMENTATION DETAILS

We implement our method using DINOv2-Base and DINOv2-Large (Oquab et al., 2024) as backbone networks. The backbone parameters are fully frozen during training, and only the parameters within our dynamic adapter modules, inserted into the last $L_a$ FFN layers, are trained. For query embedding generation, we employ $N$ learnable queries within an efficient transformer-based aggregator as used in EDTFormer (Jin et al., 2025).

Our models are trained using the AdamW optimizer with a learning rate of $2 \times 10^{-4}$ and a batch size of 128. Each batch comprises 128 locations $\times 4$ images, totaling 512 images, and the model is trained for 36 epochs. Further details regarding network architecture, hyperparameters, and training configurations are provided in the Supplementary Materials (A) to facilitate reproducibility.

## 4.3 QUANTITATIVE AND QUALITATIVE RESULTS

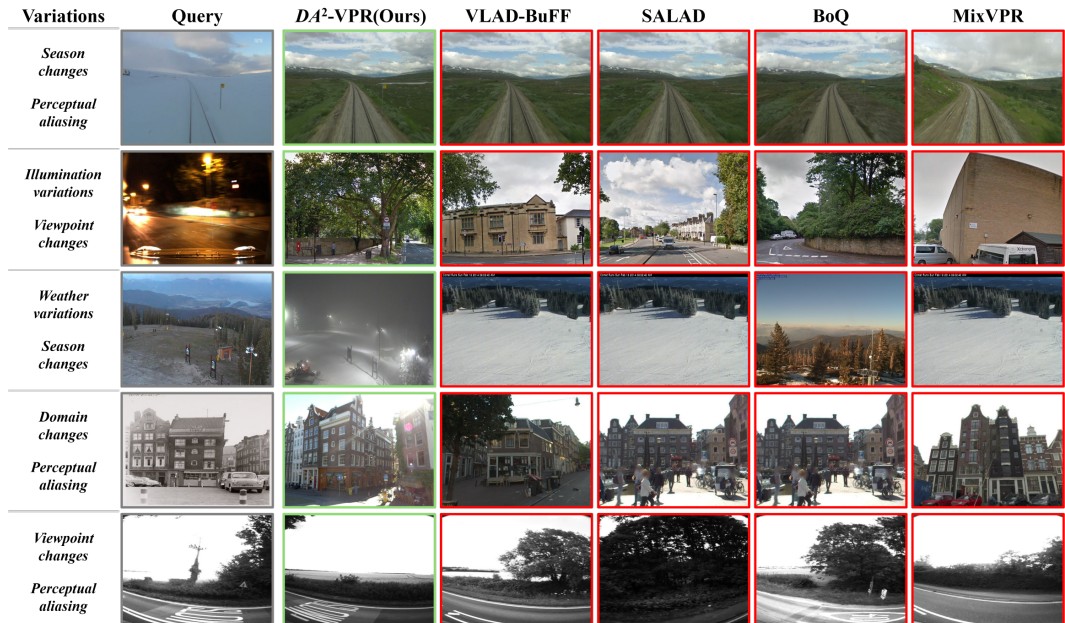

Figure 3: **Qualitative comparison under challenging domain shifts.** $DA^2$-VPR consistently retrieves correct references across diverse variations, including seasonal changes, illumination, weather, viewpoint, and domain shifts, demonstrating robustness and reliable generalization.

We quantitatively evaluate our proposed $DA^2$-VPR model against existing state-of-the-art VPR methods on diverse benchmark datasets using Recall@k (R@k) as the standard evaluation metric. We follow the evaluation protocol of MixVPR (Ali-Bey et al., 2023), considering a retrieval correct if the query and reference are within 25 meters (or within 10 frames for the Nordland* dataset).

Table 1 summarizes results on datasets with strong visual domain variations, such as significant illumination changes, seasonal variations, and extreme weather conditions. Our proposed model consistently outperformed all baseline methods across all datasets and metrics. In particular, on the Nordland dataset, which includes extreme seasonal changes and high visual similarity between scenes, our method achieves substantial improvements over other methods with both DINOv2-B and DINOv2-L backbones, clearly demonstrating the effectiveness of our dynamic feature adaptation modules under significant domain shifts.

Table 2 further presents our method's performance on standard VPR benchmark datasets, which emphasize generalization capabilities rather than variation-specific robustness. Our model consis-

Table 3: **Ablation on Dynamic Adapter (DA), Query Generation (QG), and Augmentation.** Each module contributes to performance improvement, with the full $DA^2$-VPR achieving the best results. Back. denotes trainable backbone parameters: partial-tuned (last two layers unfrozen) when DA is not applied, and frozen when DA is applied with adaptation through DA modules. Agg. denotes trainable aggregator parameters. S.N. denotes SVOX Night.

| DA | QG | Aug. | Params. (M) | | Inference time | Pitts250k | | Nordland** | | MSLS-val | | S. N. |
|----|----|------|-------------|------|----------------|-----------|------|------------|------|----------|------|-------|
| | | | Back. | Agg. | (ms) | R@1 | R@5 | R@1 | R@5 | R@1 | R@5 | R@1 |
| | | | 14.18 | 10.2 | 10.1 | 95.6 | 98.7 | 87.2 | 94.5 | 92.2 | 96.4 | 96.0 |
| ✓ | | | 0.23 | 10.2 | 11.0 | 95.8 | **98.9** | 91.8 | 97.0 | 92.2 | 96.4 | 96.2 |
| | ✓ | | 14.18 | 10.8 | 10.3 | 96.1 | 98.8 | 89.5 | 96.0 | 92.3 | 96.5 | 97.6 |
| ✓ | ✓ | | 0.23 | 10.8 | 11.2 | **96.2** | 98.8 | 92.0 | 96.9 | 92.4 | 96.4 | 97.6 |
| ✓ | ✓ | ✓ | 0.23 | 10.8 | 11.2 | **96.2** | **98.9** | **93.3** | **97.7** | **93.2** | **96.8** | **97.7** |

tently achieved state-of-the-art or highly competitive performance, demonstrating the generalization ability of our adaptive feature modulation strategy.

Qualitative results presented in Figure 3 highlight the superior robustness of our $DA^2$-VPR in challenging conditions, such as night-time illumination, extreme seasonal variations, and historical color shifts. These qualitative results further validate the effectiveness of our dynamic adaptive mechanism in modulating spatial and channel-wise features to achieve accurate place recognition under diverse visual variations.

## 4.4 ABLATION STUDY ON $DA^2$-VPR MODULES

We perform an ablation study to evaluate the contribution of each module within our proposed framework. As shown in Table 3, the baseline is established using partially tuned backbone layers paired with an efficient transformer decoder, without any dynamic modules or augmentations. Integrating the Dynamic Adapter (DA) into the baseline notably improves performance on standard datasets such as Pitts250k, as well as on challenging datasets including Nordland**, MSLS-val, and SVOX-Night (S.N.), while freezing the backbone and requiring only a small number of trainable parameters. Subsequently, adding the Reweighted Query Generation (QG) module further enhances accuracy by adaptively modulating queries based on spatial and channel-wise relevance. Lastly, incorporating synthetic augmentations inspired by RobustSAM (Chen et al., 2024) yields additional performance gains.

While the main paper focuses on the overall impact of these modules, further analyses, such as different strategies for dynamic adapter integration, the effect of varying the number of learnable queries, the impact of the number of dynamic adapter layers, and ablations on transformer decoder blocks, are provided in the Supplementary Material (C).

## 5 CONCLUSION

In this paper, we proposed $DA^2$-VPR, a dynamic visual place recognition architecture designed to address significant domain variation in real-world scenarios. By integrating dynamic adapters into the feature extraction backbone, our approach adaptively modulates intermediate representations based on input-specific visual information. Additionally, we introduced a reweighted query generation module that dynamically modulates each learnable query based on spatial relevance and channel importance. Extensive experiments conducted across diverse VPR benchmarks demonstrate the effectiveness and generalization capabilities of our proposed method. Notably, $DA^2$-VPR consistently outperforms existing models on challenging datasets such as Nordland, AmsterTime, and SVOX, validating its robustness to substantial domain shifts. However, we also observe failure cases under extreme conditions, such as severe motion blur or overly strong illumination where structural details become indistinguishable. Visualizations and detailed analyses of these cases are provided in the Supplementary Material (D.2). In future work, we plan to explore multimodal approaches that integrate complementary sensors such as LiDAR to generate more robust features, thereby enabling more accurate and resilient place recognition in challenging scenarios.

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

# SUPPLEMENTARY MATERIAL

This supplementary material provides additional details to support the main paper. It includes: (1) implementation details of each module proposed in $DA^2$-VPR, (2) descriptions of the datasets used in our experiments, and (3) extended ablation studies to further validate the effectiveness of our approach.

## A  IMPLEMENTATION DETAILS

Table 4: Configuration settings used in the experiments

| Config | Value |
|---|---|
| Precision | 16-mixed |
| Optimizer | AdamW |
| Learning rate | $2e^{-4}$ |
| Weight decay | $1e^{-3}$ |
| Batch size | 512 |
| Places | 128 |
| Images per place | 4 |
| Image size | $224 \times 224$ |
| Number of patches (L) | $16 \times 16$ |
| Patch size | $14 \times 14$ |
| Number of epochs | 36 |

The implementation details are reported in Table 4. All experiments are conducted on a NVIDIA RTX 3090 GPU using Pytorch. Training a VPR model based on the ViT-B takes 6 minutes per epoch, and requires 12 G GPU memory. Our dynamic filter employs a kernel size of $k = 3$, adapters are inserted into two transformer layers, and the decoder is composed of two blocks.

### A.1  FILTER NORMALIZATION

Filter normalization (FN) is applied to dynamically generated kernels in the dynamic adapter, inspired by Dynamic Decoupled Filter (DDF) (Zhou et al., 2021). Since the raw values of dynamically generated filters can be extremely large or small depending on the input features, directly applying them in convolution often leads to unstable training. To address this, we normalize the filters as follows:

$$D_i^{sp} = \alpha^{sp} \frac{\hat{D}_i^{sp} - \mu(\hat{D}_i^{sp})}{\delta(\hat{D}_i^{sp})} + \beta^{sp}, \ D_r^{ch} = \alpha_r^{ch} \frac{\hat{D}_r^{ch} - \mu(\hat{D}_r^{ch})}{\delta(\hat{D}_r^{ch})} + \beta_r^{ch}$$

where $\hat{D}_i^{sp}, \hat{D}_r^{ch} \in \mathbb{R}^{k \times k}$ are the spatial and channel filters before normalization, $\mu(\cdot)$ and $\delta(\cdot)$ denote the mean and standard deviation, and $\alpha, \beta$ are learnable affine parameters analogous to those in Batch Normalization (BN) (Ioffe & Szegedy, 2015).

This normalization constrains the filter values within a stable range, preventing gradient vanishing or exploding and thereby improving convergence during training.

Table 5: Summary of datasets used in evaluation

| Dataset | # query | # reference |
|---------|---------|-------------|
| Nordland* | 2760 | 27.6k |
| Nordland** | 27.6k | 27.6k |
| AmsterTime | 1231 | 1231 |
| Eynsham | 24k | 24k |
| SVOX | 14.3k | 17.2k |
| Pitts250k | 8.2k | 84k |
| Tokyo24/7 | 315 | 76.0k |
| MSLS-val | 740 | 18.9k |
| SPED | 607 | 607 |

## B  DATASET

To comprehensively evaluate domain generalization, we conduct experiments on 10 VPR benchmark datasets: Nordland*, Nordland**, AmsterTime, Eynsham, Tokyo24/7, SPED, MSLS-val, Pitts250k, and SVOX (night, snow). These datasets cover diverse visual variations, including seasonal, illumination, weather, viewpoint, and color discrepancies, providing a realistic assessment of VPR system robustness.

For training, we use GSV-Cities (Ali-bey et al., 2022), a large-scale urban dataset based on Google Street View. It includes diverse viewpoints and environmental conditions, making it a popular choice for recent VPR models aiming for robust representations.

- **Nordland** (Olid et al., 2018) provides seasonally aligned video sequences captured along a single 728 km Norwegian railway, exhibiting drastic appearance shifts from lush green summer to snow-blanketed winter. In the standard *Nordland\** protocol, 2,760 uniformly sampled summer frames act as *queries* against the full winter sequence ($\approx$27,600 frames) as the *reference*, with a strict ground-truth tolerance of only $\pm$1 frame, thereby evaluating fine-grained robustness under extreme seasonal change. In contrast, the *Nordland\*\** protocol employs the entire winter sequence ($\approx$27,600 frames) as *queries* against the full summer sequence ($\approx$27,600 frames) as the *reference*, where a looser $\pm$10-frame window ($\approx$25 m) defines correctness, enabling large-scale evaluation of scalability and retrieval consistency.

- **Amstertime** (Yildiz et al., 2022) contains challenging query-reference pairs with significant temporal gaps. Queries are grayscale historical images, while references are modern color images. Ground-truth correspondences are manually aligned.

- **Eynsham** (Cummins & Newman, 2011) is collected from a vehicle-mounted camera in low-texture rural areas in the UK. Due to repeated scenes and weak textures (e.g., fields, trees), it is challenging. Ground-truth is based on frame-level correspondences along the driving route.

- **SVOX** (Berton et al., 2021) evaluates robustness to weather-based domain shifts. It includes images of the same locations under varying conditions such as night and snow.

- **Pitts250k** (Torii et al., 2013) is a widely-used VPR benchmark with accurate GPS metadata. A fixed query set is matched against a high-resolution reference database.

- **Tokyo24/7** (Torii et al., 2015) contains query images from Tokyo captured at different times of day (day, sunset, night), making it a challenging benchmark for visual place recognition under severe illumination changes.

- **MSLS-val** (Warburg et al., 2020) consists of images from global cities with wide domain gaps, including viewpoint, heading, and distance variations. Queries and references are captured from different cameras, and a match is considered correct if within 25m and 40° angular difference.

- **SPED** (Chen et al., 2017) contains diverse CCTV views across multiple scenes, enabling the evaluation of scene-level generalization.

For SPED, Pitts250k, Tokyo24/7 and SVOX, a match is correct if the reference lies within 25 m. For MSLS-val, the threshold is 25 m and $\leq 40°$ heading difference.

Evaluation is performed using cosine similarity-based nearest neighbor retrieval. We report Recall@K (K=1, 5, 10), where a query is considered successfully localized if any of the top-K retrieved references satisfy the dataset-specific ground-truth condition.

All datasets and evaluation protocols follow the standardized settings in the visual geo-localization benchmark (Berton et al., 2022).

## C  ABLATION STUDY

### C.1  ABLATION ON DYNAMIC ADAPTER INTEGRATION STRATEGIES

Table 6: Ablation study on different adapter integration strategies. We compare (i) Frozen DINOv2 without adaptation, serial adapter insertion, and parallel adapter insertion. In the table, S.N. denotes SVOX Night.

| Ablated versions | Pitts250k | | Nordland** | | MSLS-val | | S. N. |
|---|---|---|---|---|---|---|---|
| | R@1 | R@5 | R@1 | R@5 | R@1 | R@5 | R@1 |
| Frozen DINOv2 | 95.8 | 98.7 | 83.1 | 92.3 | 92.0 | 96.2 | 97.0 |
| $DA^2$-VPR (w/ serial DA) | 96.0 | 98.8 | **94.5** | **98.2** | 92.7 | 96.5 | 97.2 |
| $DA^2$-VPR (w/ parallel DA) | **96.2** | **98.9** | 93.3 | 97.7 | **93.2** | **96.8** | **97.7** |

To evaluate the contribution of adapter integration in our framework, we conducted an ablation study with three settings: (1) Frozen DINOv2, where the backbone is used without any adaptation, (2) Serial Adapter, where the adapter is inserted sequentially after the transformer block, and (3) Parallel Adapter, where the adapter branch is integrated in parallel to the main block.

As shown in Table 6,, using adapters consistently improves performance compared to the frozen backbone. In particular, the Serial Adapter yields significant gains on datasets such as Nordland, where severe seasonal variations are present. Meanwhile, the Parallel Adapter achieves the best overall performance across most datasets. These results demonstrate the importance of adapter integration and suggest that the parallel design provides a more effective balance between stability and adaptability in diverse environments.

### C.2  EFFECT OF LEARNABLE QUERY NUMBER

Table 7: Ablation results analyzing the influence of the number of learnable queries ($Q$). performance generally improves as $Q$ increases, with diminishing returns observed after $Q = 64$, which provides an optimal balance. In the table, S.N. denotes SVOX Night.

| Size of Q | Pitts250k | | Nordland** | | MSLS-val | | S. N. |
|---|---|---|---|---|---|---|---|
| | R@1 | R@5 | R@1 | R@5 | R@1 | R@5 | R@1 |
| 8 | 95.7 | 98.8 | 80.8 | 90.9 | 91.6 | 96.1 | 95.9 |
| 16 | 95.9 | 98.7 | 90.7 | 96.3 | 91.6 | 96.2 | 96.7 |
| 32 | 96.0 | 98.8 | 92.4 | 97.2 | 92.7 | 96.2 | 97.1 |
| **64** | 96.2 | **98.9** | **93.3** | **97.7** | **93.2** | **96.8** | **97.7** |
| 96 | 96.1 | **98.9** | 92.7 | 97.2 | 92.7 | 96.5 | 97.5 |
| 128 | **96.4** | 98.8 | **93.3** | 97.6 | 92.7 | 96.5 | 97.5 |

As shown in Table 7, the model achieves competitive performance even with a small number of learnable queries ($Q$). However, increasing $Q$ generally improves retrieval accuracy, particularly in terms of $R@1$ and $R@5$, with the best performance observed at $Q = 64$. Further increasing $Q$ to 128 yields marginal gains or even slight degradation, likely due to redundancy and increased complexity. Thus, we adopt $Q = 64$ as it offers the optimal trade-off between accuracy and efficiency.

Table 8: Ablation on the number of dynamic adapter layers. param. denotes the number of trainable parameters of the adapters, while the backbone remains frozen. Only the parameters of the adapters, determined by their number, are reported. Two layers provide the best balance between performance and complexity. In the table, S.N. denotes SVOX Night.

| Num of Adapter | Params (M) | Pitts250k R@1 | Pitts250k R@5 | Nordland** R@1 | Nordland** R@5 | MSLS-val R@1 | MSLS-val R@5 | S. N. R@1 |
|---|---|---|---|---|---|---|---|---|
| 1 | 0.12 | 96.1 | **98.9** | 92.7 | 97.4 | **93.4** | **96.8** | 96.8 |
| **2** | 0.23 | **96.2** | **98.9** | **93.3** | **97.7** | 93.2 | **96.8** | **97.7** |
| 3 | 0.35 | **96.2** | 98.7 | 91.4 | 96.5 | 92.3 | 96.6 | 96.5 |
| 4 | 0.47 | 95.5 | 98.2 | 88.2 | 94.6 | 91.5 | 96.0 | 94.8 |

Table 9: Ablation study of our method on four datasets. Each module consistently improves performance over the baseline, and their combination achieves the best results. In the table, S.N. denotes SVOX Night.

| Num of decoder | Params (M) | Pitts250k R@1 | Pitts250k R@5 | Nordland** R@1 | Nordland** R@5 | MSLS-val R@1 | MSLS-val R@5 | S. N. R@1 |
|---|---|---|---|---|---|---|---|---|
| 1 | 6.1 | **96.4** | **98.9** | 92.5 | 97.2 | 92.8 | 96.6 | **97.7** |
| **2** | 10.8 | 96.2 | **98.9** | 93.3 | 97.7 | **93.2** | **96.8** | **97.7** |
| 3 | 15.5 | 96.1 | **98.9** | 93.5 | 97.8 | 92.7 | 96.5 | 97.3 |
| 4 | 20.2 | 96.1 | 98.8 | **93.9** | **97.9** | 93.1 | 96.6 | 97.6 |

## C.3 EFFECT OF DYNAMIC ADAPTER LAYERS

We analyze the effect of varying the number of Dynamic Adapter layers inserted into the backbone transformer layers (Table 8). Increasing adapters from one to two layers consistently enhances performance, suggesting that introducing additional adaptive capacity benefits feature modulation. However, further increasing adapters, beyond two results in slightly degraded performance, likely due to overfitting or optimization challenges. Hence, we select two adapter layers as the optimal configuration, balancing model complexity and performance.

## C.4 EFFECT OF TRANSFORMER DECODER BLOCKS NUMBER

As shown in Table 9, we evaluate the impact of varying the number of decoder blocks in the aggregator module. Increasing the number of decoders from one to two yields a consistent and significant improvement in both $R@1$ and $R@5$ across all datasets. However, adding more decoders (three or four) results in diminishing returns, with only marginal gains—or even slight drops in performance—despite increased model complexity. Based on this trade-off, we adopt two decoder blocks as the optimal setting.

# D VISUALIZATION

## D.1 VISUALIZATION OF ATTENTION MAP

As illustrated in Fig. 4, we compare the query attention maps of the baseline and our $DA^2$-VPR. The baseline model (partial-tuned backbone without DA, QG, or Aug., as in Table 3) often attends to unstable regions, including dynamic objects and noisy backgrounds. In contrast, our method consistently focuses on condition-invariant regions such as buildings, poles, and terrain, and even captures fine-grained structures like road markings and power lines. Moreover, the robustness of our learnable queries is evident: they remain effective under challenging conditions such as glare (first column), severe noise (third column), and grayscale inputs (fourth column). These results demonstrate that $DA^2$-VPR reliably extracts stable and discriminative cues that are crucial for robust place recognition.

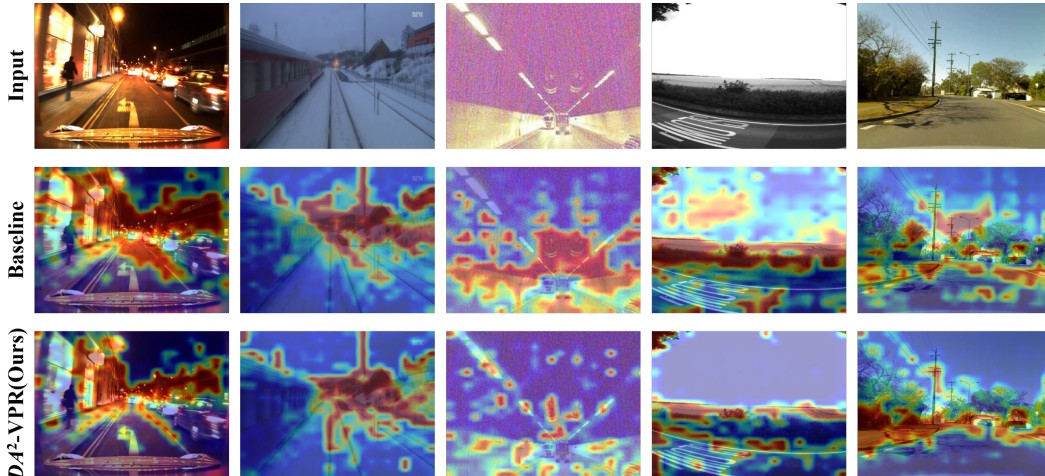

Figure 4: **Visualization of query attention maps.** The top row shows input images, the middle row illustrates the baseline model (partial-tuned backbone without DA, QG, or Aug., as in Table 3), and the bottom row presents our. Our learnable queries consistently focus on condition-invariant regions such as buildings, structures, and terrain, while ignoring dynamic elements (e.g., pedestrians, cars, trains) and noisy backgrounds, remaining robust under challenging conditions (e.g., glare, noise, grayscale).

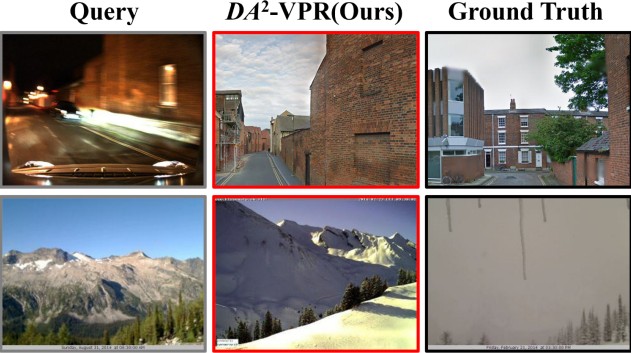

Figure 5: **Visualization of failure case.** Example queries where $DA^2$-VPR encounters challenging scenarios. The first row illustrates failure under severe motion blur and strong illumination, where essential structural cues are suppressed. The second row shows failure under extreme weather with dense fog, leading to perceptual aliasing in which visually similar but semantically different references are retrieved.

## D.2 VISUALIZATION OF FAILURE CASE

$DA^2$-VPR consistently exhibits robustness across diverse domain shifts, including illumination, seasonal, and viewpoint variations. However, as shown in Figure 5, there remain challenging cases where retrieval accuracy is affected. In the first case, severe motion blur suppresses structural cues that are essential for reliable recognition. In the second case, extreme weather with dense fog leads to perceptual aliasing, where visually similar but semantically different references are retrieved. Such observations complement our results by clarifying the boundaries of robustness and suggesting directions for further improvement.

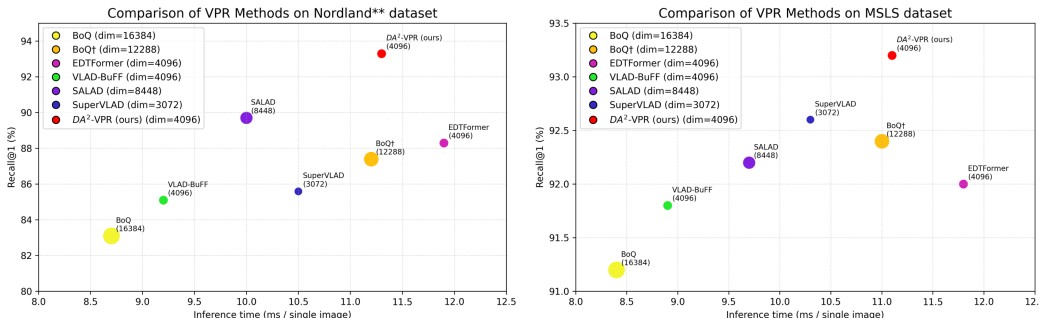

Figure 6: **Comparison of VPR methods on Nordland\*\* and MSLS datasets.** Each point represents a model, where the horizontal axis denotes inference time and the vertical axis denotes retrieval accuracy. Marker size is proportional to the descriptor dimension, providing an indication of memory and computational cost.

## E    EFFICIENCY–ACCURACY TRADE-OFF IN VPR METHODS

Figure 6 compares recent VPR methods in terms of retrieval accuracy (R@1), inference efficiency, and descriptor dimension on the Nordland\*\* and MSLS datasets. Among all baselines, our proposed $DA^2$-VPR achieves the best balance, attaining the highest accuracy and competitive inference time. These results demonstrate that $DA^2$-VPR offers a favorable trade-off between efficiency, accuracy, and representation size, making it highly suitable for practical large scale VPR deployment.

## F    EFFECT OF DOMAIN-VARIANCE AUGMENTATION

Table 10: Effect of domain-variance augmentation. the best performance for each model is highlighted in **bold**. In the table, S.N. denotes SVOX Night.

| Ablated versions | Pitts250k | | Nordland** | | MSLS-val | | S. N. |
|---|---|---|---|---|---|---|---|
| | R@1 | R@5 | R@1 | R@5 | R@1 | R@5 | R@1 |
| BoQ[†] w/o Aug. | **96.0** | **98.9** | 87.4 | **94.8** | 92.4 | **96.2** | 96.5 |
| BoQ[†] w/ Aug. | 95.8 | 98.8 | **87.7** | **94.8** | **92.7** | 95.8 | 96.5 |
| EDTFormer w/o Aug. | **95.7** | 98.5 | 85.5 | 93.7 | 92.2 | **96.6** | 95.4 |
| EDTFormer w/ Aug. | **95.7** | 98.7 | **86.8** | **93.9** | **92.7** | 96.5 | **96.0** |

As shown in Figure 10, we evaluate the impact of domain-variance augmentation by training each model under identical settings, with and without the augmentation. All models use DINOv2-B as the backbone and are trained with inputs resized to $224 \times 224$. The results indicate that domain-variance augmentation does not consistently improve performance across different models.

## G    COMPARISON WITH MODELS UNDER DIFFERENT SETTINGS

Table 11 provides a comprehensive comparison between single-stage methods using a Cross-image Encoder (w/CE) and conventional two-stage approaches.

The Cross-image Encoder(Lu et al., 2024b;d) jointly processes multiple images within a batch, allowing them to share contextual information, rather than treating each image independently. This enables images taken from different viewpoints to complement each other, where missing details in one view can be recovered by another. As a result, it shows notable improvements on datasets like Pitts250k (Torii et al., 2013), which contain multiple views of the same location. However, in practical applications, multiple images of the same place are not always available at inference time, making this approach less suitable in real-world scenarios.

Table 11: Comparison of recall@k (%) on multiple benchmark datasets with 2-stage and 1-stage (w/ cross-image encoder) models. in the table, CE. denotes cross-image encoder. the best is highlighted in **bold** and the second best is underlined, and "–" indicates values not reported. In the table, S.N. denotes SVOX Night.

| Method | Type | Nordland** | | | Pitts250k | | | MSLS-val | | | S. N. |
|--------|------|------|------|-------|------|------|-------|------|------|-------|------|
| | | R@1 | R@5 | R@10 | R@1 | R@5 | R@10 | R@1 | R@5 | R@10 | R@1 |
| CricaVPR | 1-stage | 90.7 | 96.3 | 97.6 | **97.5** | **99.4** | **99.7** | 90.0 | 95.4 | 96.4 | 85.1 |
| SuperVLAD | w/ CE | 91.0 | 96.4 | 97.7 | 97.2 | **99.4** | **99.7** | 92.2 | 96.6 | 97.4 | 94.2 |
| TransVPR | | 58.8 | 75.0 | 78.7 | 88.8 | 94.2 | 95.2 | 86.8 | 92.1 | 92.4 | - |
| R2Former | | 77.0 | 89.0 | 91.9 | 93.1 | 97.4 | 98.4 | 89.7 | 95.0 | 96.2 | - |
| SelaVPR | 2-stage | 85.2 | 95.5 | 98.5 | 95.7 | 98.8 | 99.2 | 90.8 | 96.4 | 97.2 | 89.1 |
| EffoVPR | | 95.0 | - | - | - | - | - | 92.8 | 97.2 | 97.4 | 97.4 |
| FoL | | 92.6 | 97.0 | 98.1 | 97.0 | 99.2 | 99.5 | **93.5** | 96.9 | 97.6 | **98.8** |
| $DA^2$-VPR-B | 1-stage | 93.5 | 97.6 | 98.6 | 96.2 | 98.9 | 99.4 | 93.2 | 96.8 | 96.9 | 97.7 |
| $DA^2$-VPR-L | | **95.4** | **98.4** | **99.1** | 97.0 | 99.3 | 99.5 | **93.5** | 97.3 | 97.7 | 98.5 |

Two-stage methods(Wang et al., 2022a; Zhu et al., 2023; Lu et al., 2024c; Tzachor et al., 2025; Wang et al., 2025) consist of a global retrieval step followed by a re-ranking step, allowing fast candidate selection and subsequent refinement using local features. While this structure achieves high accuracy in large-scale settings, it comes with additional computational costs and increased system complexity, limiting its suitability for real-time deployment.

Our model, by contrast, retains the efficiency of a single-stage architecture, while achieving performance comparable to or better than two-stage methods, making it both practical and effective.

## STATEMENT ON THE USE OF LARGE LANGUAGE MODELS

In the interest of transparency and in compliance with the ICLR 2026 guidelines, we report that a large language model (LLM) was used to assist in the refinement of this paper's text.

**Scope of Use.**   The model's role was strictly limited to that of a writing assistant. Its contributions include:

- Correcting grammatical errors, spelling, and punctuation.
- Improving sentence structure and flow for enhanced clarity.
- Refining word choices for greater precision and conciseness.

