# OpenReview forum: "$DA^2$-VPR: Dynamic Architecture for Domain-Aware Visual Place Recognition"
_ICLR.cc/2026/Conference — ICLR 2026 Conference Withdrawn Submission_

### Official Review · Reviewer_gGEV · 2025-10-30

**Soundness:** 2
**Presentation:** 3
**Contribution:** 2
**Rating:** 2
**Confidence:** 5

**Summary:**

The paper proposes DA2-VPR, a dynamic feature modulation framework for VPR that adapts features based on scene characteristics. It introduces: A Dynamic Adapter for spatial and channel-wise modulation of features in a frozen DINOv2 backbone. A Reweighted Query Generator that adjusts transformer queries according to spatial and channel relevance. A Domain-variance augmentation strategy for robustness to environmental changes.

**Strengths:**

addresses domain shift, a known challenge in VPR.

evaluates across diverse benchmarks and provides ablations.

outperforms baselines like BoQ and EDTFormer on multiple datasets.

**Weaknesses:**

1. The novelty claim (“first to dynamically modulate VPR features”) seems overstated given EMVP and conditional query transformers already address similar dynamics. The dynamic adapter + query weighting combination is incremental, essentially applying known ideas to VPR with minimal architectural innovation.

2. The training and test setups mostly follow standard VPR splits. There’s no explicit out-of-distribution experiment proving domain robustness. Ablation tables are limited to component removal; there’s no analysis of parameter sensitivity, training stability, or runtime under domain variations.

3. The motivation around domain awareness is vague. There’s no explicit mechanism to identify or characterize domains. It’s merely feature reweighting based on input embeddings.

4. The domain-variance augmentation part is superficially described and lacks quantitative impact evaluation.

**Questions:**

None.

---

### Official Review · Reviewer_pbzk · 2025-10-31

**Soundness:** 2
**Presentation:** 3
**Contribution:** 2
**Rating:** 4
**Confidence:** 3

**Summary:**

The paper introduces DA2-VPR, a dynamic architecture for domain-aware visual place recognition. The main challenge addressed is the performance degradation of VPR systems when encountering domain shifts such as changes in lighting, season, or viewpoint between training and test environments. DA2-VPR proposes a dynamic feature modulation framework that adapts its representations based on the input scene characteristics. This dynamic adaptation is achieved through three key components: (1) a dynamic adapter that adjusts features across spatial and channel dimensions, (2) a transformer-based aggregator that generates adaptive query embeddings, and (3) a domain-variance augmentation strategy. The approach is tested on several VPR benchmarks with significant domain shifts, achieving superior performance compared to traditional input-invariant methods.

**Strengths:**

1. The paper proposes a dynamic feature modulation mechanism for VPR, which is an important step forward compared to static adaptation techniques. The idea of adjusting representations based on scene-specific features makes the method more robust and adaptable.
2. The approach is tested on multiple challenging datasets that cover diverse environmental variations such as lighting, seasonal changes, and viewpoint differences. The model consistently outperforms existing methods, demonstrating its effectiveness in real-world scenarios.

**Weaknesses:**

1. Although the paper proposes a novel Dynamic Architecture for Domain-Aware Visual Place Recognition (DA2-VPR), the motivations behind key design choices are not sufficiently detailed. For instance, while the method integrates a dynamic adapter into the feature extraction backbone, the rationale for choosing this specific architecture over other potential solutions (e.g., different forms of dynamic adaptation or alternative feature extraction architectures) is not clearly justified. A deeper explanation of why this specific integration works better in addressing domain variation compared to simpler approaches (e.g., static adapters or fine-tuning layers) would strengthen the paper's contribution. The authors could discuss in more detail how the dynamic modulations on spatial and channel dimensions enhance generalization.
2.The idea of dynamically modulating features based on the input's visual characteristics is promising, but the paper does not provide sufficient theoretical underpinnings or analysis of how these modulations contribute to improved performance. The authors mention that the dynamic adapter helps in dealing with domain shifts, but a deeper exploration of how this mechanism interacts with different environmental conditions (lighting, viewpoint, weather, etc.) would clarify the adaptive process. More quantitative analysis, perhaps through ablation studies or theoretical modeling, would better showcase the impact of these modulations.
3. The ablation study primarily evaluates the contributions of the dynamic adapter, query generation, and augmentation. However, it is not clear how each component of the DA2-VPR system compares to existing methods that also aim to handle domain shifts. For example, comparing the proposed dynamic feature modulation against simpler methods, such as classic fine-tuning or hybrid approaches like those in EMVP, could highlight the unique advantages of the DA2-VPR architecture. Additionally, the impact of different configuration choices, such as the number of learnable queries or the number of layers in the dynamic adapter, could be expanded further.
4. While the paper mentions that the DA2-VPR performs well on challenging datasets, it only briefly touches on failure cases under extreme conditions like motion blur and strong illumination. A more thorough analysis of these failure cases would be beneficial. It would also be helpful to compare how DA2-VPR performs in these extreme scenarios against baseline models, detailing whether the failure is intrinsic to the model's design or could be mitigated with further modifications (e.g., multi-modal sensors like LiDAR or additional training strategies).

**Questions:**

1. The paper introduces a dynamic adapter integrated into the feature extraction backbone, but the motivation for this design choice is not clearly explained. Could you provide more detailed reasoning behind selecting this specific architecture over alternative methods like static adapters or fine-tuning layers? What unique benefits does the dynamic adapter offer in terms of addressing domain variations, and why is it more effective than simpler approaches? Additionally, how do the spatial and channel dimension modulations contribute specifically to improving the model s generalization capabilities across varying environmental conditions (e.g., lighting, season, viewpoint)?
2. The concept of dynamically modulating features based on the visual characteristics of the input is interesting, but the paper lacks a thorough theoretical explanation of how these modulations contribute to performance improvements. Could you provide more detailed analysis or a theoretical justification for the dynamic modulation mechanism? Specifically, how does this mechanism adapt to different environmental shifts, such as changes in lighting, weather, or viewpoint, and why does this lead to better generalization compared to existing methods?
3.In the ablation study, the contributions of the dynamic adapter, query generation, and augmentation are evaluated. However, there is no clear comparison of how DA2-VPR performs relative to other existing methods, such as EMVP, that also aim to handle domain shifts. Could you provide a direct comparison of your method against simpler and more established techniques, such as static fine-tuning, EMVP, or other hybrid approaches? This would help highlight the unique strengths of DA2-VPR in handling domain shifts.
4. The paper briefly mentions failure cases in extreme conditions, such as motion blur and strong illumination, but does not provide a detailed analysis of these failures. Could you elaborate on these failure cases, and compare how DA2-VPR performs under these extreme scenarios in comparison to baseline models?

---

### Official Review · Reviewer_T7eR · 2025-11-01

**Soundness:** 2
**Presentation:** 3
**Contribution:** 2
**Rating:** 6
**Confidence:** 3

**Summary:**

This paper introduces a visual place recognition method with a DINOv2-based image feature extraction backbone, a dynamic adaptation module for finetuning with different conditioned scenes, a transformer to aggregate features, and some data augmentation techniques to aid training. The dynamic adapter is inspired by the dynamic decouple network, which multiplies the spatial-wise filter and the channel-wise filter before applying to the input images. The transformer is used to reweight the query generation, and the data augmentation follows the synthetic degradations with multiple weather condition augmentations. The authors have conducted various comparisons, including some ablation studies on different benchmarks, and the proposed method achieves good performance.

**Strengths:**

- A standard paper with good performance showing on various VPR benchmarks.
- The proposed method introduces the dynamic decoupled network structure into the VPR conditional adaptation for image features, and shows good improvement.
- The paper writing is standard, and the presentation is relatively good with clear logic.
- If not cherry-picking, Figure 3 has shown a really impressive result.

**Weaknesses:**

- The authors could have given more quantitative or qualitative results for different conditions to further strengthen the reported performance. For example, the authors could report quantitative results per condition changes.
- The authors have provided Figure 5 for failure cases. However, these failure scenarios are too general for VPR. It would be better for the authors to analyze using this specific method, what the biggest challenge remains to be solved, where the failures occurred, and provide more on why this is the case.
- Efficiency is another aspect to consider in more discussions in the main paper. The authors could consider moving Figure 6 to the main paper for a more complete analysis of the method.

**Questions:**

Please check the above comments for more details.

---

### Official Review · Reviewer_czq8 · 2025-11-01

**Soundness:** 2
**Presentation:** 2
**Contribution:** 2
**Rating:** 2
**Confidence:** 4

**Summary:**

The paper proposes a dynamic feature-modulation framework for VPR: dynamic adapters are inserted into the later stages of a frozen DINOv2 backbone to perform learnable modulation along two branches (spatial and channel) to handle domain shifts such as lighting, season, and weather. It further designs a query reweighting mechanism: spatial and channel weights are computed from the input features and used to adaptively reweight the learnable queries in the Transformer aggregator. The method is evaluated on multiple datasets with strong domain shifts (e.g., Nordland, SVOX night/snow) as well as standard benchmarks (Pitts250k, Tokyo24/7, SPED, MSLS).

**Strengths:**

1. The figures are clear and intuitively illustrate the proposed method.
2. The experiment is conducted on multiple standard benchmarks.

**Weaknesses:**

1. The differences between this work and existing studies has not been adequately clarified. Specifically, the dynamic module in Figure 2 of this paper is similar to the Decoupled Dynamic Filter Module in Figure 2 of the previous work [1]. Moreover, Equation (3) in this paper is identical to Equation (2) in the previous work [1].
2. The overall system architecture, incorporating a dynamic adapter, reweighted query generation, and a transformer aggregator, is more complex than simpler VPR baselines, potentially hindering deployment and reproducibility. More important, there is no experimental comparison about computational overhead and complexity.
3. This paper only provides results on the MSLS-val dataset and lacks results on the MSLS-challenge dataset. Since the labels for MSLS-challenge are not publicly available and online testing is required, results on this dataset would be more persuasive. Moreover, I would like to confirm whether the code will be full released if this paper is accepted.
4. In Table 3, it should be clearly stated that the parameters are trainable; otherwise, readers might mistakenly assume that the proposed model has very few parameters.
5. There are several minor issues in the paper. For example, in the caption of Table 1, "DINO" should be changed to "DINOv2." Additionally, the first word of the second sentence in the caption of Table 2 is not capitalized.

**Questions:**

Please see weaknesses.

---

### Note · Authors · 2025-11-14

**Comment:**

We sincerely thank the reviewers and the program committee for their time and constructive feedback. After careful consideration, we believe that our current submission is not yet suitable for ICLR. Therefore, we have decided to withdraw the paper.

**Withdrawal Confirmation:**

I have read and agree with the venue's withdrawal policy on behalf of myself and my co-authors.